# A Fast Calibration Method for Photonic Mixer Device Solid-State Array Lidars

**DOI:** 10.3390/s19040822

**Published:** 2019-02-17

**Authors:** Yayu Zhai, Ping Song, Xiaoxiao Chen

**Affiliations:** Key Laboratory of Biomimetic Robots and Systems (Ministry of Education), Beijing Institute of Technology, Beijing 100081, China; zyythinking@foxmail.com (Y.Z.); 2120160158@bit.edu.cn (X.C.)

**Keywords:** photonic mixer device, PMD solid-state array lidar, array complementary metal–oxide–semiconductor photodetector calibration, electrical analog delay method, modular lens distortion correction, pixel point adaptive interpolation

## Abstract

The photonic mixer device (PMD) solid-state array lidar, as a three-dimensional imaging technology, has attracted research attention in recent years because of its low cost, high frame rate, and high reliability. To address the disadvantages of traditional PMD solid-state array lidar calibration methods, including low calibration efficiency and accuracy, and serious human error factors, this paper first proposes a calibration method for an array complementary metal–oxide–semiconductor photodetector using a black-box calibration device and an electrical analog delay method; it then proposes a modular lens distortion correction method based on checkerboard calibration and pixel point adaptive interpolation optimization. Specifically, the ranging error source is analyzed based on the PMD solid-state array lidar imaging mechanism; the black-box calibration device is specifically designed for the calibration requirements of anti-ambient light and an echo reflection route; a dynamic distance simulation system integrating the laser emission unit, laser receiving unit, and delay control unit is designed to calibrate the photodetector echo demodulation; the checkerboard calibration method is used to correct external lens distortion in grayscale mode; and the pixel adaptive interpolation strategy is used to reduce distortion of distance images. Through analysis of the calibration process and results, the proposed method effectively reduces the calibration scene requirements and human factors, meets the needs of different users of the lens, and improves both calibration efficiency and measurement accuracy.

## 1. Introduction

Three-dimensional (3D) measurement technologies have been widely used in surveying, structural measurements, virtual reality, and unmanned driving over the past few decades [1,2,3,4,5]. Among these technologies, and unlike two-dimensional (2D)/3D mechanical scanning lidar [6,7], which requires fast rotation scanning to obtain depth data, solid-state array lidar based on time-of-flight measurements provides the metric distance to the scene from the sensor [8,9,10]; in particular, photonic mixer device (PMD) solid-state array lidar can reduce the reliability problems associated with mechanical rotating equipment and increase the frame rate while reducing the complexity of 3D reconstruction from the data.

PMD solid-state array lidar is widely used in the computer vision community. For example, it is applied to human–computer interactions that require gestures and real-time motion recognition [11,12]; simultaneous localization and mapping in robotics, unmanned vehicles, and drones [13]; and virtual reality in game-type entertainment equipment [14]. The basic composition of a PMD solid-state array lidar system is shown in Figure 1. The system emits laser light through an array laser; the emitted laser light is then reflected by obstacles and the reflected light is received by the array complementary metal–oxide–semiconductor (CMOS) photodetector. The processor then calculates the distance between the lidar system and the target [9,10]. In this process, the pulse echo signals are integrated multiple times using two different capacitors (C1, C2) and two different phase windows (Phase A, Phase B) under the same pixel; the time of the pulse flight is then calculated based on the integration results for the different capacitors, and the target distance is then finally calculated from the flight time.

However, PMD solid-state array lidar is affected by the chip temperature, chip design, lens characteristics, and distance calibration [15,16,17,18,19] and thus has low calibration efficiency, poor measurement accuracy, and other operational issues.

Several studies have been conducted on PMD solid-state array lidar calibration. Several works focused on calibration of a 3D time-of-flight (ToF) sensor system. Fuchs and Hirzinger [20] presented a calibration procedure that allowed the user to calibrate the distance-related and amplitude-related errors of a ToF camera for a desired operating range and also determined the extrinsic parameters of the ToF camera. Kuhnert and Stommel proposed a distance calibration approach [21] that centered on the corrected pixel and corrected the phase difference by linearly fitting the 5 × 5 surrounding pixels to obtain the true distance. After correction of a given input image, a min/max depth map was calculated by examining the neighbors of each pixel, which led to development of a confidence interval for the true distance information. Kahlmann et al. [22] presented a systematic calibration method that considered different influencing parameters, including the reflectivity, the integration time, the temperature, and the distance itself. These parameters were analyzed with respect to their effects on the distance-measuring pixels and their output data were determined. Lindner and Kolb [23] combined a per-pixel linear calibration with a global B-spline fit that provided improved local control and was especially useful for online calibration tasks. Schiller et al. [24] developed a calibration method that estimated the full intrinsic calibration of a photonic mixing device (PMD) camera and included consideration of both optical lens distortions and systematic range errors. Their calibration approach uses a planar checkerboard pattern as a calibration reference for multiple cameras and distances. Schmidt [25] proposed a calibration approach that performed an implicit calibration of the sensor for homogeneities in arbitrary raw data that were acquired from a scene. Jung et al. [26] presented a calibration method for a ToF sensor and color camera pair to align 3D measurements with a color image correctly. They designed a 2.5-dimensional pattern board with irregularly placed holes to be detected accurately from both the low-resolution depth images of a ToF camera and high-resolution color images. In general, these calibration systems set a reflective baffle at a specified initial distance from the PMD solid-state array lidar and then vary this distance repeatedly, thus obtaining ranging results at different distances. These ranging results are then compared with the actual distances to acquire the calibration data. However, there are several problems involved in using these systems, including the heavy workload and the inevitability of human interference because the scene needs to be set manually.

Several works have focused on the calibration algorithm used for PMD solid-state array lidar. Some studies [27,28] used a single integration time adjustment, while others [29,30] selected an appropriate integration time based on the average amplitude data of a scene; the results of these approaches are not ideal for scenarios with both a foreground and a background. Steiger et al. [31] discussed a method for setting of the optimal global integration time; however, all their distance errors were above the uncertainties specified by the manufacturer even after calibration. Swadzba et al. [32] used a stepwise optimization and particle filtering algorithm based on the 3D ToF camera. Reynolds et al. [33] proposed an improved per-pixel confidence measure using a random forest regressor that was trained using real-world data, but the confidence assignment speed was slow. Pathak et al. [34] analyzed a scenario in which a pixel beam is intersected by more than one object and where the assumption of a unimodal probabilistic distribution causes spurious objects to appear in the detected scene. The drawback of their method was the requirement for integration of over 100 images for each frame, which reduced the frame rate significantly. Lindner et al. [35] described a fast algorithmic approach to combine high-resolution RGB images with PMD distance data that were acquired using a binocular camera, but the simple threshold based on segmentation using the PMD autocorrelation amplitude does not always lead to optimal results close to object boundaries. Kim et al. [36] proposed a novel denoising algorithm for ToF depth images based on estimation of the space-varying ToF depth noise and introduced a parametric model for ToF depth noise that used the infrared intensity while assuming additive white Gaussian noise. Kern [37] proposed a calibration technique for a laser scanner using a plane with holes, but the specific objective in that case was to calibrate a laser scanner that provides much more accurate depth measurements than an array lidar. This method is thus unsuitable for array lidar systems.

Overall, most current calibration systems calibrate distance errors by placing calibration plates at different distances and most current calibration methods calibrate distance errors using complex settings or algorithms. However, some problems do exist in these systems and methods, including heavy workloads, complex computational requirements and serious human interference because of the need to set the scene manually. Therefore, based on an analysis of the ranging error of PMD solid-state array lidar systems, a calibration method for the CMOS photodetector array is proposed that uses a black-box calibration device and an electrical analog delay method. With the aim of meeting the demand for improvement of the lens field of view, a modular lens distortion correction method that uses a checkerboard and pixel point adaptive interpolation optimization approach is proposed. The main contributions are as follows:(1)From an analysis of the actual laser emission modulation signal, the echo demodulation error of the PMD solid-state array lidar system is obtained through a detailed study of the echo demodulation processes of a sinusoidal modulation wave and a rectangular modulation wave. This provides a theoretical basis for rapid calibration of PMD solid-state array lidar at long distances.(2)A black-box calibration device is specially prepared for the calibration requirements of anti-ambient light and the echo reflection route to reduce the external interference on the one hand while also improving the uniformity of the received light on the other hand.(3)A dynamic distance simulation system that integrates a laser emission unit, a laser receiving unit, a delay control unit, and other units is designed. The echo-demodulation of the CMOS photodetector array is calibrated using the electrical analog delay. The delay phase-locked loop is used to set different laser emission times to simulate different calibration distances. We correct the calibration curve linearly to improve the delay time accuracy. This method realizes rapid and long-distance range calibration for the CMOS photodetector array without changing the calibration distance.(4)A checkerboard image is captured in grayscale, and the internal parameters and distortion coefficients of the lens that are obtained using the calibration method of Zhang [38] are used to correct the distortion of the external lens. To address the problems where the distortion correction pixels do not correspond exactly and the distance image collected using the area lidar has no depth values for some pixels, the pixel adaptive interpolation strategy is used to reduce the distortion. This method can meet the needs of the different users of the lens and achieve modular calibration.

## 2. System Introduction

The calibration methods proposed in this paper include two aspects, comprising CMOS photodetector array calibration and modular lens distortion correction, as shown in Figure 2. The calibration of the CMOS photodetector array involves analysis of the sinusoidal wave–rectangular wave modulation and demodulation error, use of the automatic calibration method in combination with the black-box calibration device and the electrical analog delay; and calibration for lens distortion via lens installation, internal parameter/distortion coefficient calibration, coordinate conversion/distortion correction, and pixel point adaptive interpolation optimization. 

## 3. Materials and Methods

### 3.1. Analysis of Lidar Modulation and Demodulation Errors

The PMD solid-state array lidar system mainly uses the ToF to calculate the distance to the target. The specific working process is illustrated in Figure 3. In this process, the pulse echo signals are integrated multiple times using two different capacitors and different phase windows under the same pixel; the time of the pulse flight is then calculated based on the integration results for the different capacitors, and, finally, the target distance is calculated from the flight time.

#### 3.1.1. Sinusoidal-Wave Modulation and Demodulation

At present, the most commonly used ToF measurement method is the sinusoidal-wave laser modulation and demodulation method; in this method, a laser emits a sinusoidally modulated wave and this sinusoidally modulated wave encounters an object that reflects or scatters the light during flight; the light then returns to the CMOS photodetector array along the opposite path. The CMOS photodetector array demodulates the echo signal to obtain the laser flight time and multiplies it by the speed of light to obtain the relative distance to the target. The specific sinusoidal wave modulation and demodulation process is illustrated in Figure 4. Four different phase windows (0°, 180°, 90°, and 270°) for the two capacitors of a single pixel are used to demodulate the echo signal and obtain the phase change of the echo signal.

We define the transmitted modulated wave here as *a* + *b*sin(*ωt*). The reflected wave is weaker than the transmitted wave because of the effects of propagation and reflection but the reflected and transmitted waves have the same frequency. The reflected wave is then defined as *A* + *B*sin*ω*(*t–t_TOF_*). According to the processes of modulation and demodulation shown in Figure 4, the integral control signal controls capacitances C1 and C2 to integrate them separately. The integration results for four phase windows in a single cycle are taken here as an example:(1)$$\begin{array}{c} {}^{s}Q{}_{1}^{DC0}=\int_{0}^{\frac{\pi }{\omega }}\left[A+B\sin\omega (t-t_{TOF})\right]dt,\\ {}^{s}Q{}_{2}^{DC0}=\int_{\frac{\pi }{\omega }}^{\frac{2\pi }{\omega }}\left[A+B\sin\omega (t-t_{TOF})\right]dt,\\ {}^{s}Q{}_{1}^{DC1}=\int_{-\frac{\pi }{2\omega }}^{\frac{\pi }{2\omega }}\left[A+B\sin\omega (t-t_{TOF})\right]dt,\\ {}^{s}Q{}_{2}^{DC1}=\int_{\frac{\pi }{2\omega }}^{\frac{3\pi }{2\omega }}\left[A+B\sin\omega (t-t_{TOF})\right]dt, \end{array}$$
where ${}^{s}Q{}_{1}^{DC0}$, ${}^{s}Q{}_{2}^{DC0}$, ${}^{s}Q{}_{1}^{DC1}$, and ${}^{s}Q{}_{2}^{DC1}$ are the modulated wave conditions in the case of a sinusoidal wave, i.e., they are the amount of charge obtained by integrating C1 when the phase window is 0°, the amount of charge obtained by integrating C2 when the phase window is 180°, the amount of charge obtained by integrating C1 when the phase window is 90°, and the amount of charge obtained by integrating C2 when the phase window is 270°, respectively. *ω* is the frequency of the sinusoidal wave, *t_TOF_* is the flight duration, and the capacitance integral charge differences ${}^{s}DC0$ and ${}^{s}DC1$ based on sinusoidal wave demodulation can be obtained from Equation (1) as follows:(2)$$\begin{array}{c} {}^{s}DC0=4\times \frac{B}{\omega }\times \cos\omega t_{TOF},\\ {}^{s}DC1=-4\times \frac{B}{\omega }\times \sin\omega t_{TOF}. \end{array}$$

In addition, the flight duration is:(3)$${}^{s}t{}_{TOF}=\frac{1}{\omega }\times \left[\pi +\mathrm{atan}2\left({}^{s}DC0,\;-{}^{s}DC1\right)\right],$$
Where atan2(*x,y*) is calculated as:(4)$$\mathrm{atan}2\left(x,y\right)=\left\{ \begin{array}{ll} \mathrm{atan}(y/x) & x>0\\ \mathrm{atan}(y/x)+\pi & x<0,y\geq 0\\ \mathrm{atan}(y/x)-\pi & x<0,y<0\\ \pi /2 & x=0,y>0\\ -\pi /2 & x=0,y<0\\ 0 & x=0,y=0 \end{array}\right..$$

The CMOS photodetector array controls the time-sharing integration of capacitors C1 and C2 through the integral switch control signal to realize demodulation of the reflected wave, and the distance to the target can then be resolved based on the demodulation result. However, the sinusoidal modulation signal that is emitted by the PMD solid-state array lidar often fails to meet the ideal requirements. The actual curve is shown in Figure 5. The actual signal is similar to a rectangular wave, and this wave is low-pass filtered because of the limitations of the generator bandwidth. This paper therefore uses rectangular waves for further modulation and demodulation analyses.

#### 3.1.2. Rectangular Wave Modulation and Demodulation

The rectangular modulated wave laser beam encounters objects and is reflected or scattered during flight and then returns to the CMOS photodetector array along the opposite path. The CMOS photodetector array then demodulates the echo signal to obtain the laser flight duration and the relative distance to the target by considering the speed of light. The specific rectangular wave modulation and demodulation process is shown in Figure 6. Four different phase windows (0°, 180°, 90°, and 270°) of the two capacitors from a single pixel are used to demodulate the echo signal and thus obtain the phase change of the echo signal. The demodulation of the rectangular reflected wave is similar to the demodulation of the sinusoidal reflected wave. According to the modulation and demodulation process shown in Figure 6, the integral control signal controls capacitances C1 and C2 to integrate them separately.

The integration results ${}^{r}Q{}_{1}^{DC0}$, ${}^{r}Q{}_{2}^{DC0}$, ${}^{r}Q{}_{1}^{DC1}$, and ${}^{r}Q{}_{2}^{DC1}$ for the four phase windows in a single cycle are used as an example here:(5)$$\begin{array}{c} {}^{r}Q{}_{1}^{DC0}=\int_{0}^{\frac{\pi }{\omega }}f(t)dt,\\ {}^{r}Q{}_{2}^{DC0}=\int_{\frac{\pi }{\omega }}^{\frac{2\pi }{\omega }}f(t)dt,\\ {}^{r}Q{}_{1}^{DC1}=\int_{-\frac{\pi }{2\omega }}^{\frac{\pi }{2\omega }}f(t)dt,\\ {}^{r}Q{}_{2}^{DC1}=\int_{\frac{\pi }{2\omega }}^{\frac{3\pi }{2\omega }}f(t)dt, \end{array}$$
where *f*(*t*) is a reflected wave and is defined as: (6)$$f(t)=\left\{ \begin{array}{l} A\;t_{TOF}<t\leq t_{TOF}+\frac{\pi }{\omega }\\ 0\;t_{TOF}+\frac{\pi }{\omega }+<t\leq t_{TOF}+\frac{2\pi }{\omega } \end{array}\right..$$

From this,$\;{}^{r}Q{}_{1}^{DC0}$, ${}^{r}Q{}_{2}^{DC0}$, ${}^{r}Q{}_{1}^{DC1}$, and ${}^{r}Q{}_{2}^{DC1}$ and *t_TOF_* have the following relationships:(7)$$\begin{array}{ll} {}^{r}Q{}_{1}^{DC0}=-A(t_{TOF}-\frac{\pi }{\omega }) & 0<t_{TOF}\leq \frac{\pi }{\omega }\\ {}^{r}Q{}_{2}^{DC0}=At_{TOF} & 0<t_{TOF}\leq \frac{\pi }{\omega }\\ {}^{r}Q{}_{1}^{DC1}=\left\{ \begin{array}{ll} -A(t_{TOF}-\frac{\pi }{2\omega }) & 0<t_{TOF}\leq \frac{\pi }{2\omega }\;\\ A(t_{TOF}-\frac{\pi }{2\omega }) & \frac{\pi }{2\omega }<t_{TOF}\leq \frac{\pi }{\omega }\; \end{array}\right.\\ {}^{r}Q{}_{2}^{DC1}=\left\{ \begin{array}{ll} A(t_{TOF}+\frac{\pi }{2\omega }) & 0<t_{TOF}\leq \frac{\pi }{2\omega }\\ -A(t_{TOF}-\frac{3\pi }{2\omega }) & \frac{\pi }{2\omega }<t_{TOF}\leq \frac{\pi }{\omega } \end{array}\right. \end{array}$$

Furthermore, the capacitance integral charge differences ${}^{r}DC0$ and ${}^{r}DC1$ based on rectangular wave demodulation are:(8)$$\begin{array}{l} {}^{r}DC0={}^{r}Q{}_{2}^{DC0}-{}^{r}Q{}_{1}^{DC0}\\ {}^{r}DC1={}^{r}Q{}_{2}^{DC1}-{}^{r}Q{}_{1}^{DC1} \end{array}$$

Because the actual modulation wave is not exactly the same as a sinusoidal wave or a rectangular wave, and because the sinusoidal wave demodulation mode is simultaneously simpler and more real-time-based than the rectangular wave demodulation mode, the actual flight time is usually calculated using Equation (3). Therefore, substitution of ${}^{r}DC0$ and ${}^{r}DC1$ as obtained by rectangular wave demodulation into Equation (3) provides *^r^t_TOF_*. However, ${}^{r}DC0$ and ${}^{r}DC1$ are different from ${}^{s}DC0$ and ${}^{s}DC1$, and there is thus a difference between the calculated and real values of *^r^t_TOF_*.

### 3.2. Calibration Method

By analyzing the ranging error of PMD solid-state array lidar, this paper proposes a calibration method for a CMOS photodetector array based on a black-box calibration device and an electrical analog delay method. First, according to the calibration requirements of anti-ambient light and the echo reflection route, we construct a black-box calibration device that reduces external interference on the one hand while also improving the uniformity of the received light on the other hand. Second, a dynamic distance simulation system that integrates a laser emission unit, a laser receiving unit, a delay control unit, and various other units is designed. This system calibrates the echo demodulation of the photodetector using the electrical analog delay method and uses a delay phase-locked loop to set various laser emission times to simulate different calibration distances without actually changing the calibration distance. Simultaneously, the calibration curve is corrected linearly to improve the delay time accuracy.

#### 3.2.1. Black-box Calibration Device

A schematic diagram of the black-box calibration device is shown in Figure 7. The black-box calibration device mainly comprises six panels, a cylindrical cylinder, and a white calibration plate. The six panels and the cylindrical tube are made from black nonreflective materials. The white calibration plate is the same size as the rear panel and is fixed on the inner side of the rear panel. The front panel has two holes for the laser modulation signal emitted by the incident laser and the echo laser signal emitted back to the CMOS photodetector array. The black-box device is made entirely from black materials except for the white calibration plate.

In this work, the infrared light is reflected by the white calibration plate during the transmission process and only the directly reflected light can enter the CMOS photodetector array. All other reflected light is absorbed by the black surfaces, and this again helps to avoid the effects of light entering the CMOS photodetector array after repeated reflections from other surfaces during calibration.

#### 3.2.2. Calibration of CMOS Photodetector Array Based on the Electrical Delay Method

This paper presents a dynamic distance simulation system design that integrates a laser emission unit, a laser receiving unit, a delay control unit, and other units into a single system. A schematic diagram of the design is shown in Figure 8. The delay length that can be input in real time is realized using synchronous clock counting and a delay phase-locked loop. The time interval between the laser emission and the laser echo can thus be simulated using the delay time to allow simulation of high-precision dynamic distance information and calibration of the photodetector echo demodulation.

Figure 9 shows a schematic diagram of the delay phase-locked loop when generating different delay times. The delay time for each step is τ. The total delay is thus *n* × τ and the virtual distance range that can be simulated is [τ×c/2+a,n×τ×c/2+a], where *c* is the speed of light, *n* is an integer and *a* is the distance from the lidar to the reflective panel of the black-box calibration device. Using different values of the virtual distance *d*, the test distance d′ can be calculated, and the calibration curve of the virtual distance and the test distance can thus be obtained. The distance is corrected using this calibration curve during real-time measurements.

### 3.3. Lens Distortion Correction

The depth image is distorted and correction of the depth image is thus an essential step. We use a checkerboard to calibrate lidar systems. After the lidar system’s internal parameters and distortion coefficients are obtained, we can correct the image distortion. We need to modify the traditional correction algorithm because the distortion correction pixels do not correspond exactly and the depth image collected using the ToF lidar has no depth values for some pixels. The calibration process is presented in Figure 10. 

The specific steps are as follows:(1)Install the lens for different angles of view at the front end of the PMD solid-state array lidar CMOS photodetector array.(2)Print a checkerboard grid to act as a target and fix it on a hard sheet.(3)Change the relative positions of the PMD solid-state array lidar and the target and acquire multiple images of the target from different angles in the grayscale mode of the lidar system.(4)Extract the feature points from each image and select the corner points of the checkerboard to act as calibration points.(5)Find the plane projection matrix H for each image.(6)Solve for the internal parameters of the PMD solid-state array lidar system using the matrix H.(7)Optimize the calibration results by back-projection transformation to obtain more accurate calibration results and calculate the distortion coefficients of the PMD solid-state array lidar system.(8)Use the internal parameters of the PMD solid-state array lidar system to convert the normalized plane points and the pixel plane points.(9)Correct the distortion of the PMD solid-state array lidar system using the distortion coefficients.(10)Process the depth image pixel points using a pixel adaptive interpolation strategy.

Step 1 is installation of the lens, which allows the lens to be selected by different users with different needs and also solves the limitations of using a single-field-angle lens. Steps 2–7 use the checkerboard and the calibration method of Zhang that was mentioned earlier to acquire the internal parameters and the distortion coefficients of the PMD solid-state array lidar system, which are not detailed in this paper. Steps 8–10 are the processes of coordinate conversion, distortion correction, and pixel adaptive interpolation, and are described as follows. 

Steps 8–9: Depth information of the pixel in the corrected image is obtained in the following steps, as illustrated in Figure 11.

First, we project the pixel in the corrected image onto the normalized plane using the internal parameters of the PMD solid-state array lidar system:(9)$$\left[\begin{array}{c} u\\ v\\ 1 \end{array}\right]=\left[\begin{array}{ccc} f_{x} & 0 & c_{x}\\ 0 & f_{y} & c_{y}\\ 0 & 0 & 1 \end{array}\right]\left[\begin{array}{c} x\\ y\\ 1 \end{array}\right]$$

Here, (*u*,*v*,1) denotes the pixel’s coordinates in the pixel coordinate system, while (*x*,*y*,1) denotes the coordinates in the normalized coordinate system. *f_x_*, *f_y_*, *c_x_*, and *c_y_* are the internal lidar parameters, where *f_x_* and *f_y_* denote the focal lengths of the lidar system in the *x* and *y* directions, respectively, and (*c_x_*,*c_y_*) denotes the coordinates of the principal point in the image coordinate system.

Second, we correct the distortion of the depth image:(10)$$\begin{array}{c} x_{correct}=x\left(1+k_{1}r^{2}+k_{2}r^{4}\right)+2p_{1}xy+p_{2}\left(r^{2}+2x^{2}\right)\\ y_{correct}=y\left(1+k_{1}r^{2}+k_{2}r^{4}\right)+p_{1}\left(r^{2}+2y^{2}\right)+2p_{2}xy \end{array}$$
where (*x_correct_*, *y_correct_*, 1) denotes the coordinates of the corrected point in the pixel coordinate system. $r=\;\sqrt{x^{2}+y^{2}}$, *k*_1_, *k*_2_, *p*_1_, and *p*_2_ are all distortion coefficients.

Third, we project the corrected pixel in the normalized plane to the original image using the internal lidar parameters.

Step 10: Because the distortion correction pixels do not correspond exactly and the depth image acquired using the ToF lidar has no depth values for some pixels, we must modify the traditional correction algorithm. We interpolate the depth values of the pixels. The pixel adaptive interpolation strategies for the different cases are presented in Table 1.

In the table, a green point represents the projection of corrected pixels on the uncorrected image. Purple pixels represent the points closest to the green point. Solid purple pixels have depth information, while hollow pixels have no depth information. *D_0_*, *D_1_*, *D_2_*, *D_3_*, and *D_4_* are the distances to the center point, the lower-left pixel, the top-left pixel, the lower-right pixel and the top-right pixel, respectively. (*x_p_*_0_,*y_p_*_0_), (*x_p_*_1_,*y_p_*_1_), (*x_p_*_2_,*y_p_*_2_), (*x_p_*_3_,*y_p_*_3_), and (*x_p_*_4_,*y_p_*_4_) are the coordinates of the center point, the lower-left pixel, the top-left pixel, the lower-right pixel, and the top right pixel, respectively. In addition, α_x_ = *x_p_*_0_ − *x_p_*_1_, α_y_ = *y_p_*_0_ − *y_p_*_1_, and (*m*,*n*) denotes the coordinates of the center point in the barycentric coordinate system.

## 4. Experiments and Results

### 4.1. Results of Array CMOS Photoelectric Sensor Calibration

#### 4.1.1. Results of Theoretical Error Analysis

According to the theoretical analysis, the relationship between the flight time obtained by sinusoidal wave–rectangular wave modulation and demodulation and the real flight time is given by:(11)$${}^{r}t{}_{TOF}=\left\{ \begin{array}{ll} \frac{1}{\omega }\times \left[\pi +\mathrm{atan}2\left(2At_{TOF}-A\frac{\pi }{\omega },\;-2At_{TOF}\right)\right] & \;0<t_{TOF}\leq \frac{\pi }{2\omega }\\ \frac{1}{\omega }\times \left[\pi +\mathrm{atan}2\left(2At_{TOF}-A\frac{\pi }{\omega },\;2At_{TOF}-2A\frac{\pi }{\omega }\right)\right] & \frac{\pi }{2\omega }<t_{TOF}\leq \frac{\pi }{\omega } \end{array}\right..$$

In this paper, the flight time error is simulated using MatLab. Figure 12 shows the flight time simulation error obtained under the assumption that *f* = 10 MHz.

In the PMD solid-state array lidar calibration process, the value of *t_TOF_* is (*n* × 2 + *t*_0_) ns, where *n* is an integer. For comparison with the actual calibration process, the theoretical correction results for the PMD solid-state array lidar at modulation frequencies of 12 MHz and 24 MHz are shown in Figure 13.

#### 4.1.2. Actual Experimental Results

This paper verifies the calibration method for the CMOS photodetector array on the PMD solid-state array lidar platform. The experimental setup is shown in Figure 14. 

The equipment in the figure is partially blocked because the PMD solid-state array lidar system is still at the research and development stage. Among the system parameters, the laser emission intensity is set before calibration and the calibration begins when the echo intensity that is observed on the display interface is moderate and the CMOS photodetector array is receiving uniform light. The reflected signal intensity is moderate and consistent during the calibration process. In addition, to avoid noise, the distance is only calculated when the reflected signal intensity reaches a specific value in the actual measurement process, so the signal intensity is not taken into account during the error simulation. The actual calibration curves are shown in Figure 15a and Figure 16a. The comparison between theoretical analysis results and actual calibration results are shown in Figure 15b and Figure 16b. Figure 15 shows the distance calibration results obtained when the modulation frequency is 12 MHz. The phase offset is related to the distance to the black-box calibration device itself. Figure 16 shows the distance calibration results obtained when the modulation frequency is 24 MHz. The offset is again related to the distance to the black-box calibration device itself. The truncated position indicates the farthest distance at which the corresponding virtual distance exceeded the 24 MHz modulation frequency when the number of delay-phase-locked loop steps is 30.

### 4.2. Modular Lens Distortion Calibration Results

This work verifies the modular lens distortion calibration method on the PMD solid-state array lidar platform. The experimental device used is shown in Figure 17.

The checkerboards used for the PMD solid-state array lidar system in grayscale mode are shown in Figure 18.

The lens internal parameters and the distortion coefficients are listed in Table 2.

Figure 19 presents three experimental results that were obtained after distortion correction and pixel adaptive interpolation. The scene is an office environment with dimensions of 10 m × 10 m × 3 m and the illumination is approximately 500 lux. The CMOS photodetector array described in this paper responds to the optical band range of 600–900 nm, so the CMOS photodetector array will collect ambient light (i.e., sunlight and other light) and emit optical signals simultaneously during the actual measurement process. However, the modulation and demodulation method proposed in this paper cancels out the common ambient light via multi-capacitance synchronous charge accumulation, which means that the measurement error is reduced. In addition, the PMD solid-state array lidar takes approximately 40 μs to output a single frame of data. 

### 4.3. Distance Test Results After Completion

The accuracy of the PMD solid-state array lidar measurements is evaluated following the calibration process. The experimental platform is shown in Figure 20, where the actual distance is obtained using a high-precision single-point laser range finder (LEICA DISTO A6, Heerbrugg, St. Gallen, Switzerland). The measurement accuracy is evaluated by comparing the distances to the center point of the white plate obtained when measured using the LEICA DISTO A6 and the PMD solid-state array lidar system. 

After the LEICA DISTO A6 and the PMD solid-state array lidar have been moved continuously, the distances to the center point of the white plate measured using the LEICA DISTO A6 are compared with the same distances when measured using the PMD solid-state array lidar over the 0.5–5 m range; the results are presented in Figure 21. During the actual measurement process, the LEICA DISTO A6 was used as the standard. The reason why the LEICA DISTO A6’s curve does not appear to be straight is that we did not measure it at the same proportional distance.

### 4.4. Performance Comparison

We have compared the performance of our calibration method with that of Steiger et al. [31], Lindner et al. [28], Schiller et al. [24] and Jung et al. [26]. The results of this comparison in terms of distance error, calibration time and scene setup are given in Table 3, where the term “NaN” is used to indicate instances where no metrical data were available at some of the distances.

The results in Table 3 indicate that the method proposed in this paper is superior to the methods described in [28,31] but is slightly poorer than the methods described in [24,26]. However, the proposed method is superior to all other methods in terms of both calibration time and scene setup. Our method only requires 10 min to perform the complete calibration process, including setup of the calibration scene, initialization and calculation, while the other methods all require dozens of minutes at a rough estimate. At the same time, in terms of the required scene scope, the requirements of our study are much smaller than those of the other methods. More importantly, our scene scope will not change with calibration distance.

## 5. Discussion

We have presented a fast calibration method for PMD solid-state array lidar. First, based on an analysis of the PMD solid-state array lidar ranging error, we have proposed a calibration method for the CMOS photodetector array based on a black-box calibration device and an electrical analog delay method. Second, we presented a lens distortion correction method based on a checkerboard and pixel adaptive interpolation to resolve the limitations of use of the single-field-angle lens. We highlight the following findings of the study:(1)By analyzing the actual laser emission modulation signals, the echo demodulation error of the PMD solid-state array lidar system was obtained based on a detailed study of the echo demodulation process of a sinusoidal modulation wave and a rectangular modulation wave. This provided a theoretical basis for rapid calibration of PMD solid-state array lidar over a large distance range. Comparison of Figure 13, Figure 14, Figure 15 and Figure 16 indicates that the theoretical error analysis results are basically consistent with the actual error analysis results, which further verifies the correctness of the theoretical analysis.(2)As shown in Figure 14, this paper presents the design of a dynamic distance simulation system that integrates a laser emission unit, a laser receiving unit, a delay control unit, and various other units. Echo demodulation of the CMOS photodetector array was performed using the black-box calibration device and the electrical analog delay method. Table 3 shows that, in terms of accuracy, the method provided by our group is superior to those proposed in [28,31] but is slightly poorer than those presented in [24,26]. However, the proposed method is superior to all other methods in terms of both calibration time and scene setup. The apparatus and the method can calibrate a CMOS photodetector array over a large distance range without changing the calibration distance.(3)Different users have different requirements for the angle of view of the lens. We have proposed a modular lens distortion correction method based on a checkerboard. Figure 18 and Table 2 show that the internal parameters and the distortion coefficients of the lens can be obtained using the calibration method of Zhang and these parameters can then be used to correct the lens distortion. To address the problems in that the distortion correction pixels do not correspond exactly and the distance images collected via area lidar have no depth values for some pixels, this paper proposed a pixel adaptive interpolation strategy to achieve distortion optimization. Figure 19 shows that the method can correct distortion for the different angles of view.

## 6. Conclusions

To address the disadvantages of the traditional PMD solid-state array lidar calibration method, which include low calibration efficiency, low calibration accuracy, and serious human factors, this paper proposed a calibration method for a CMOS photodetector array based on a black-box calibration device and an electrical analog delay method. By analyzing the error performance in demodulation of a sinusoidal modulation wave and a rectangular modulation wave, a dynamic distance simulation system was then designed by integrating a laser emission unit, a laser receiving unit, a delay control unit, and other units into a single system. The photodetector echo demodulation was calibrated using the electrical analog delay method, which verified the correctness of the theoretical results. This method effectively reduces the calibration scene requirements, human factors and material requirements while improving the calibration efficiency. In addition, a modular lens distortion correction scheme based on a checkerboard and pixel adaptive interpolation was proposed. In the grayscale mode of the PMD solid-state array lidar system, the calibration method of Zhang was used to analyze a checkerboard image, and the external lens internal parameters and distortion coefficients were obtained to correct the lens distortion. Simultaneously, we adopted a pixel point adaptive interpolation strategy to reduce the distortion. The proposed method meets the needs of the different users of the lens and improves the universality of PMD solid-state array lidar technology.

## Figures and Tables

**Figure 1 sensors-19-00822-f001:**
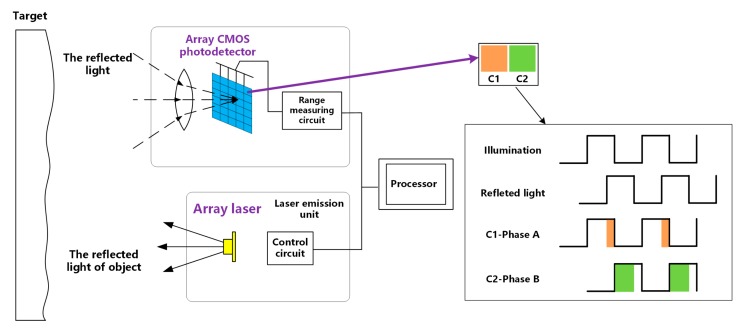
Composition of PMD solid-state array lidar system.

**Figure 2 sensors-19-00822-f002:**
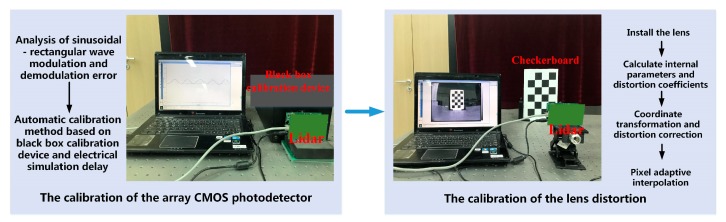
System overview.

**Figure 3 sensors-19-00822-f003:**
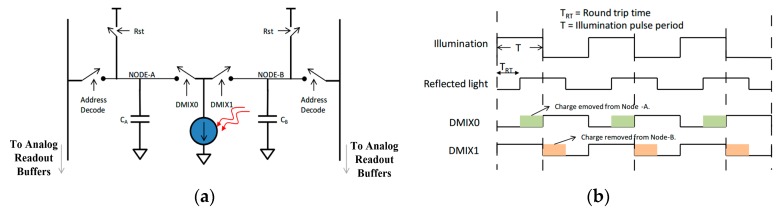
Specific working processes of the PMD solid-state array lidar system. (**a**) The different capacitor charging processes of the same pixel. (**b**) Charge integration processes for different phase windows.

**Figure 4 sensors-19-00822-f004:**
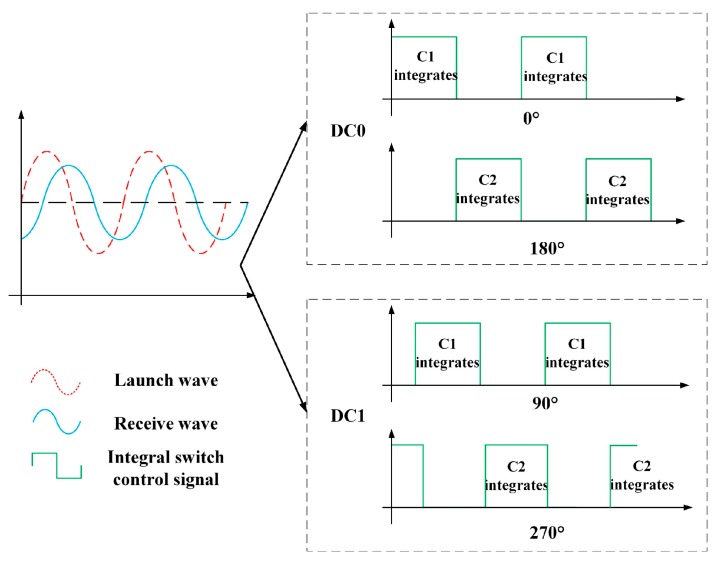
Sinusoidal-wave modulation and demodulation processes.

**Figure 5 sensors-19-00822-f005:**
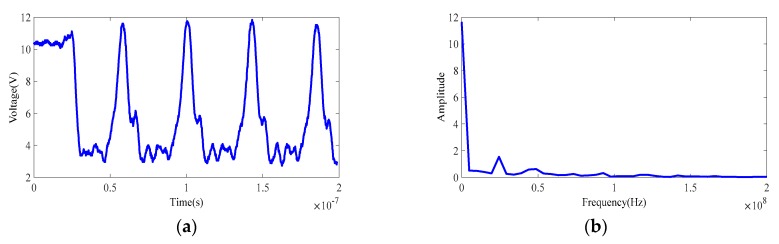
Actual sinusoidal modulation signal and its Fourier transform. (**a**) Actual sinusoidal modulation signal. (**b**) Fast Fourier transform of the sinusoidal modulation signal.

**Figure 6 sensors-19-00822-f006:**
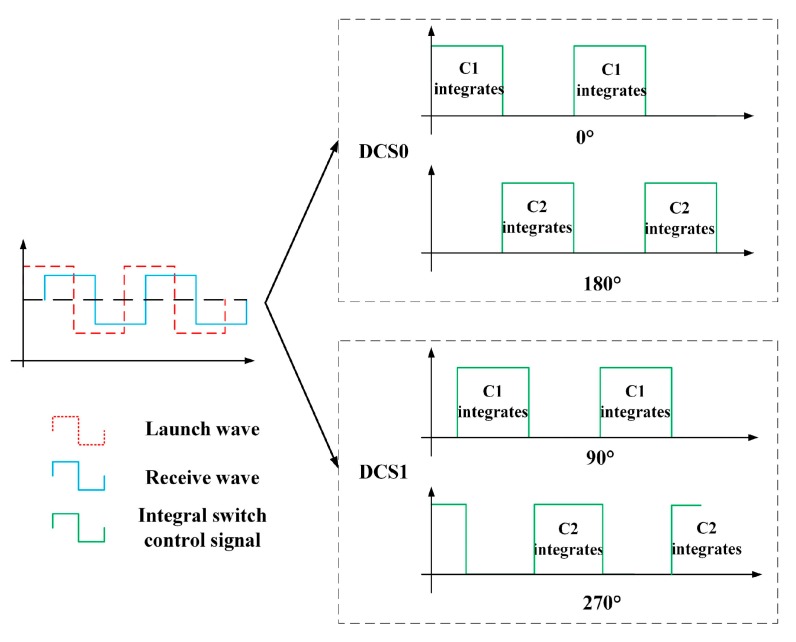
Rectangular wave modulation and demodulation process.

**Figure 7 sensors-19-00822-f007:**
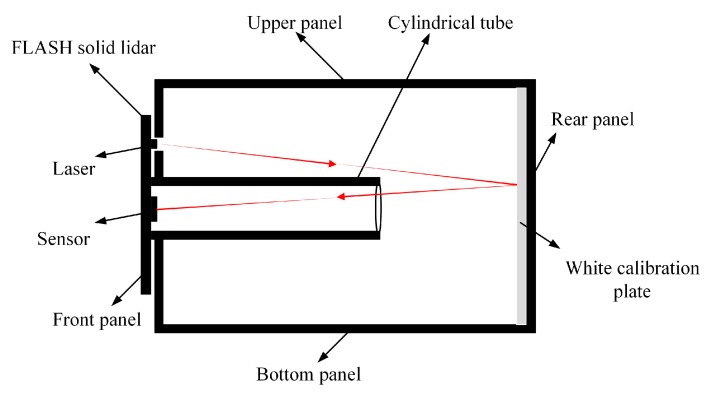
Schematic of the black-box calibration device.

**Figure 8 sensors-19-00822-f008:**
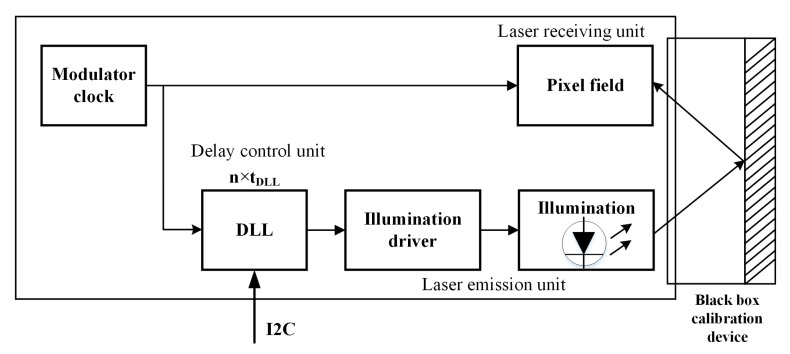
Calibration diagram of CMOS photodetector array based on the electrical delay method.

**Figure 9 sensors-19-00822-f009:**
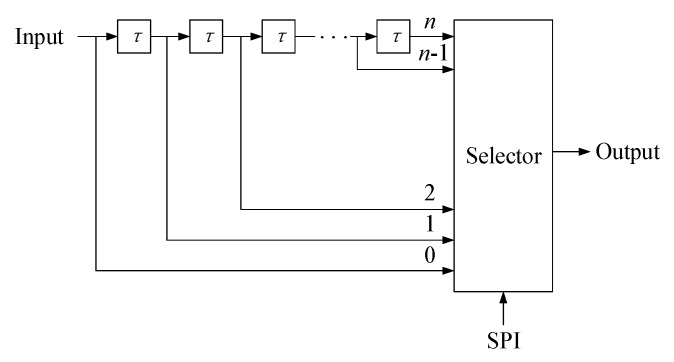
Delayed phase-locked loop when generating different delays.

**Figure 10 sensors-19-00822-f010:**
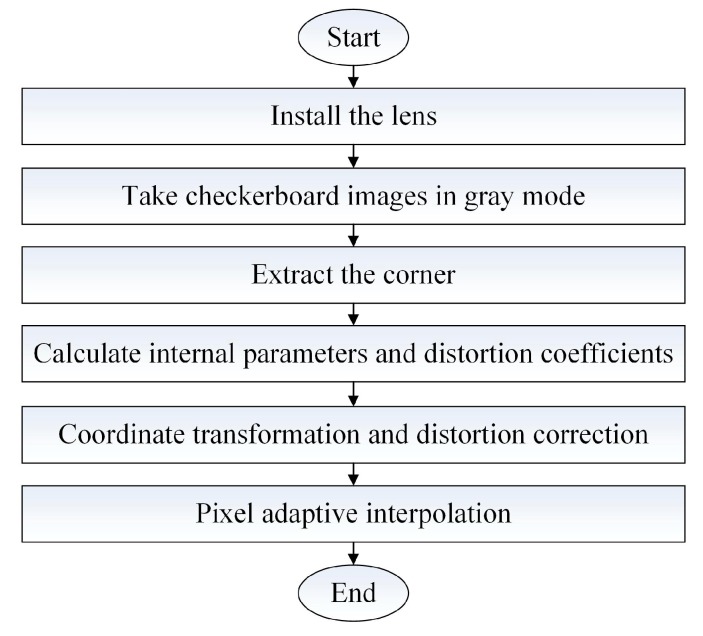
Flowchart for the lens distortion correction process for PMD solid-state array lidar.

**Figure 11 sensors-19-00822-f011:**
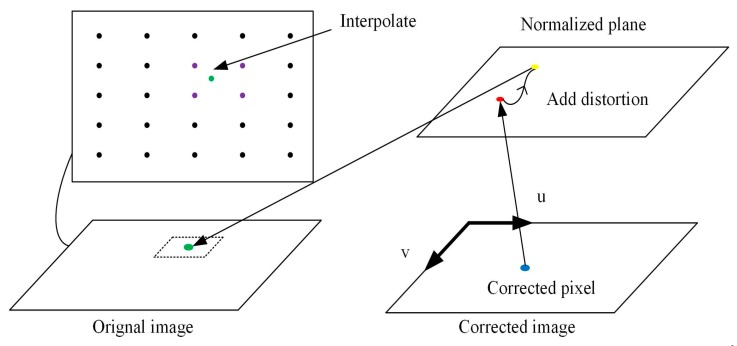
Schematic diagram of correction of the depth image.

**Figure 12 sensors-19-00822-f012:**
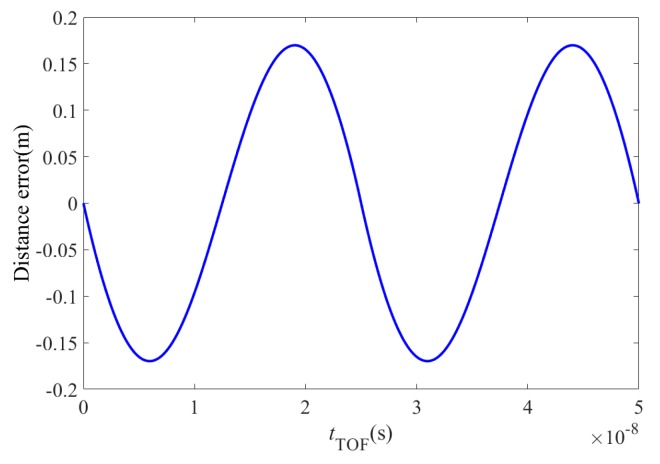
Error in the flight time calculation.

**Figure 13 sensors-19-00822-f013:**
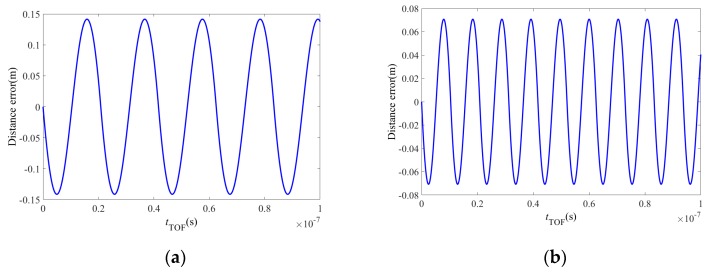
Theoretical error analysis of calibration results at modulation frequencies of 12 MHz and 24 MHz. (**a**) Modulation frequency of 12 MHz. (**b**) Modulation frequency of 24 MHz.

**Figure 14 sensors-19-00822-f014:**
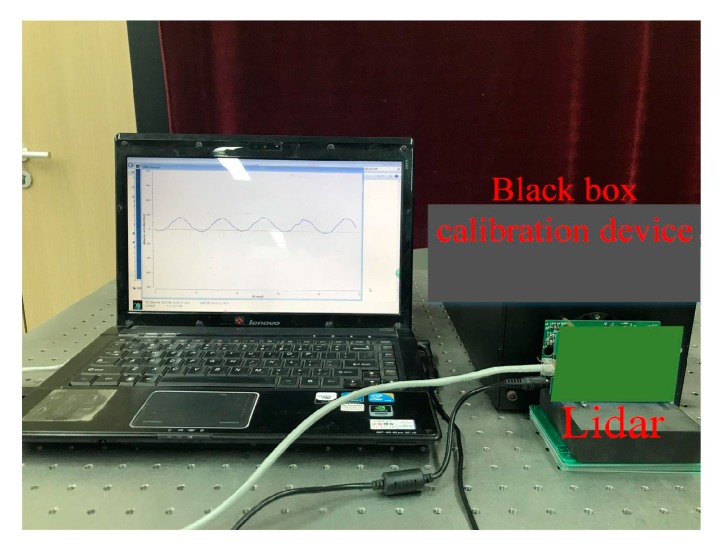
Calibration system for the CMOS photoelectric sensor array of the PMD solid-state array lidar system.

**Figure 15 sensors-19-00822-f015:**
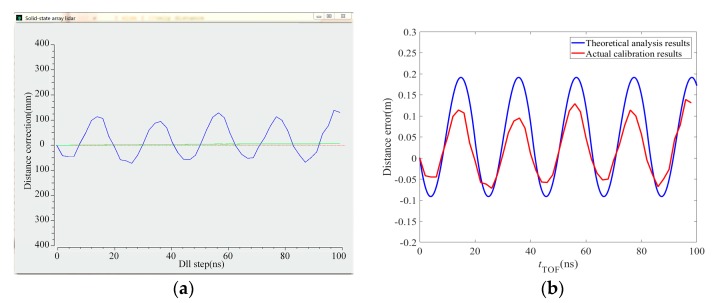
Distance calibration results when the modulation frequency was 12 MHz. (**a**) The calibration curve. (**b**) Comparison between theoretical analysis results and actual calibration results.

**Figure 16 sensors-19-00822-f016:**
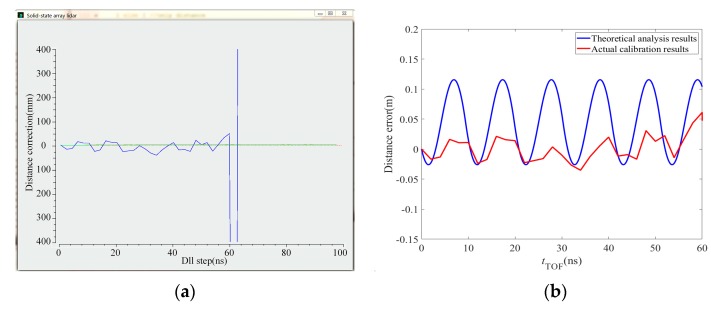
Distance calibration results when the modulation frequency was 24 MHz. (**a**) The calibration curve. (**b**) Comparison between theoretical analysis results and actual calibration results.

**Figure 17 sensors-19-00822-f017:**
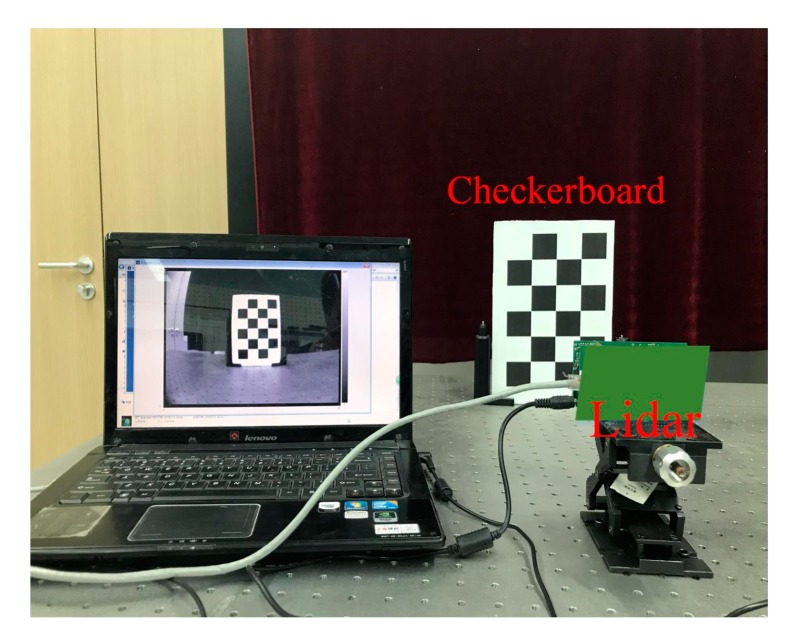
Experimental device used for modular lens distortion calibration.

**Figure 18 sensors-19-00822-f018:**
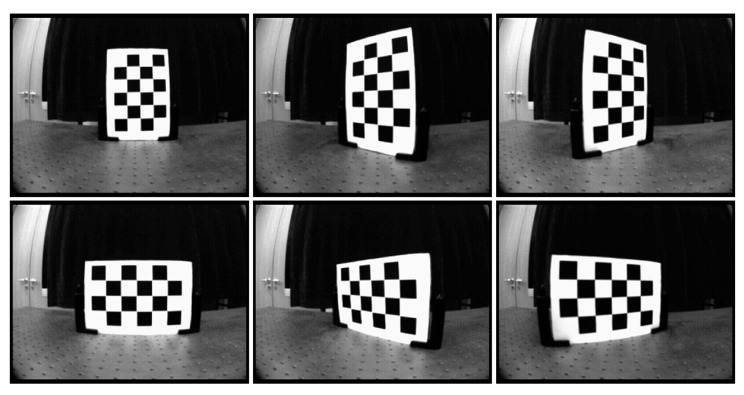
Grayscale images of the partial checkerboard patterns.

**Figure 19 sensors-19-00822-f019:**
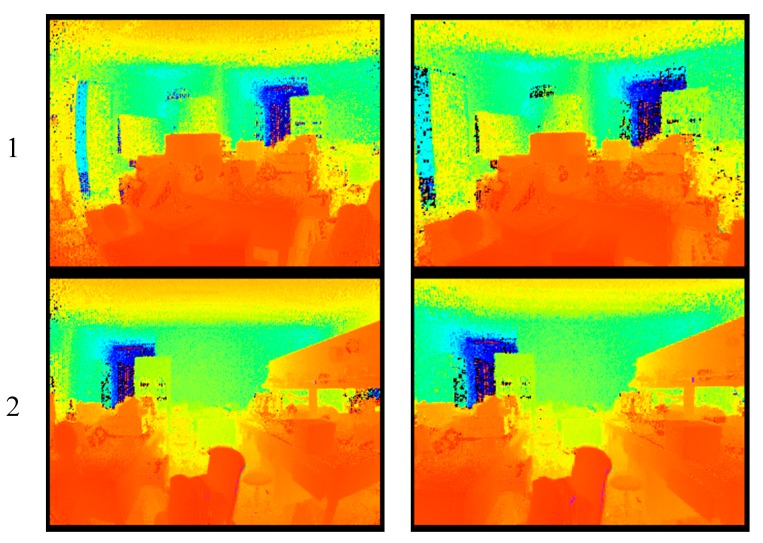
Three experimental results obtained after distortion correction and pixel adaptive interpolation. (**a**) Distance images without distortion correction. (**b**) Distance images after distortion correction.

**Figure 20 sensors-19-00822-f020:**
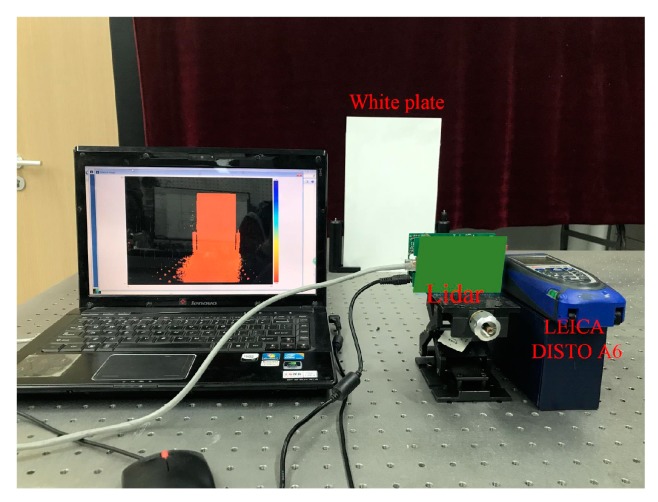
Distance test platform.

**Figure 21 sensors-19-00822-f021:**
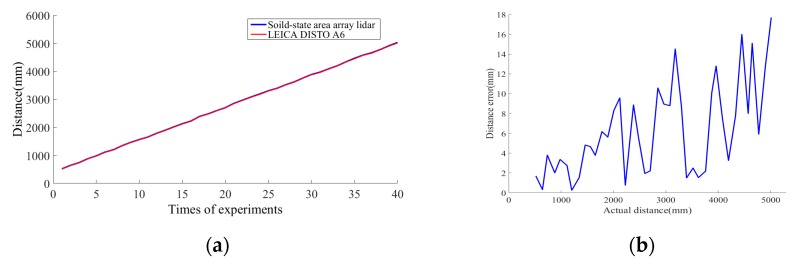
Distance measurement results. (**a**) Comparison of distance measurements made using the LEICA DISTO A6 and the PMD solid-state array lidar system. (**b**) Errors in the distance measurement results.

**Table 1 sensors-19-00822-t001:** Pixel adaptive interpolation strategies for different cases.

Case	Pixel Adaptive Interpolation Strategy
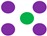	D0=(1−ax)×(1−ay)×D1+(1−ax)×ay×D2+ax×(1−ay)×D3+ax×ay×D4
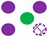	D0=mD1+nD2+(1−m−n)D4
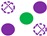	D0=(1−xp0+yp02)×D1+xp0+yp02×D4
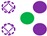	D0=ayD4+(1−ay)×D3
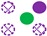	D0=Dx
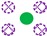	D0=NaN

**Table 2 sensors-19-00822-t002:** Lens internal parameters and distortion coefficients.

**Internal Parameters**	*f_x_*	*f_y_*	*c_x_*	*c_y_*
	207.767	209.308	174.585	129.201
**Distortion Coefficients**	*k* _1_	*k* _2_	*p* _1_	*p* _2_
	−0.37568	0.15729	0.00304	0.00046

**Table 3 sensors-19-00822-t003:** Performance comparison.

	Distance Error (mm)										Calibration Time	Scene Scope
	Test Distance	900	1100	1300	1700	2100	2500	3000	3500	4000		
**Steiger et al. [31]**		NaN	3 (at 1207)		25 (at 1608)	57 (at 2250)	NaN	NaN	NaN	NaN	About dozens of minutes	Not mentioned
**Lindner et al. [28]**		19.4	28.2	21.0	28.9	13.5	17.3	15.9	21.8	26.7	About dozens of minutes	About 4 m × 0.6 m × 0.4 m
**Schiller et al. [24] (automatic feature detection)**		7.45 (mean)							NaN	NaN	About dozens of minutes	About 3 m × 0.6 m × 0.4 m
**Schiller et al. [24] (some manual feature selection)**		7.51 (mean)							NaN	NaN	About dozens of minutes	About 3 m × 0.6 m × 0.4 m
**Jung et al. [26]**		7.18 (mean)							NaN	NaN	137 s (calculation) About dozens of minutes (scene setup)	About 3 m × 0.6 m ×1 m
**Our method**		3.37	4.82	6.17	8.30	9.57	8.88	12.79	10.58	14.52	90 s (calculation) 10 min (calculation, scene setup and initialization)	About 0.8 m × 0.4 m × 0.3 m

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
