# Peer review of "A Fast Calibration Method for Photonic Mixer Device Solid-State Array Lidars"

_sensors, 2019, doi:10.3390/s19040822_

Reviewer 1 Report

1. Usually a flash LiDAR consists of a direct TOF camera which measures range in one shot and it does not have motion blur. Your system is an indirect TOF camera which captures at least two frames for range calculation. It is not a flash and it has motion blur.

 2. The intention of the black box test is good. However, a black surface does not necessarily look black under NIR source.

 3. For calibration process using DLL delay, figure 15 and 16 should include theoretical curve for comparison. DLL step in ns should be included in text and figures.

4. Figure 21, the x-axis should be actual distance but not times of experiments. And it is not clear to me why LEICA DISTO A6's curve is not linear. If it is used as a reference, its curve should be straight.

 The two calibration methods are both good. The expected results and the actual measurement results should be put in the same graph for comparison.

Author Response

Dear Editors:

    Thank you very much for your letter and for the reviewers’ comments concerning our manuscript entitled “Fast Calibration Method for FLASH Solid-state Array Lidar” (# sensors-424552). These comments are valuable and have been very helpful for revising and improving our paper, as well as providing important guiding significance for our research. We have revised the manuscript according to the reviewers’ comments to make the paper more acceptable. In order to make the changes easily viewable for you and the reviewers, the revisions have been clearly highlighted using red font in Microsoft Word and PDF. This cover letter explains, point by point, the details of our revisions in the manuscript and our responses to the reviewers’ comments.

        The attachment is the detailed content.

 Thank you.

Dr. Yayu Zhai

Beijing Institute of Technology

[email protected]

Reviewer 2 Report

The authors proposed a practical calibration procedure to improve the performance of 3D flash lidar cameras. Though lacking novelty in the proposed methods, the detailed works presented in this manuscript could provide helps to the progress of the ToF camera applications.

My major concern is that the manuscript didn’t provide enough background knowledge before going into the details. LiDAR is a research topic of a wide range with many different schemes, therefore it would be nice if the author could provide a concise but clear explanation of how the flash LiDAR camera works. Figure 1 didn’t have enough information for that purpose, for example, the principle of range measurement is not given, for those who are not familiar with flash LiDAR cameras, this would be confusing. Furthermore, I am not following the reason why the word ‘flash’ is capital throughout the manuscript, usually people capital only the acronyms. This also causes confusion to readers.

There are other ToF based 3D ranging techniques without using raster scan and don’t have a computational overhead, such as “Single-pixel three-dimensional imaging with time-based depth resolution, Nature Communications, 7, 12010, (2016)”, which I think deserves referring, given the context.

The using of English language is not grammatically wrong in most cases but somehow improper, I think proofreading is in order.

 I would recommend the manuscript for publication in Sensors, providing my concerns could be addressed properly.

Author Response

Dear Editors:

     Thank you very much for your letter and for the reviewers’ comments concerning our manuscript entitled “Fast Calibration Method for FLASH Solid-state Array Lidar” (# sensors-424552). These comments are valuable and have been very helpful for revising and improving our paper, as well as providing important guiding significance for our research. We have revised the manuscript according to the reviewers’ comments to make the paper more acceptable. In order to make the changes easily viewable for you and the reviewers, the revisions have been clearly highlighted using red font in Microsoft Word and PDF. This cover letter explains, point by point, the details of our revisions in the manuscript and our responses to the reviewers’ comments.

        The attachment is the detailed content.

 Thank you.

Dr. Yayu Zhai

Beijing Institute of Technology

[email protected]

Round  2

Reviewer 2 Report

I think the manuscript has been improved to provide more information for general readers.

I agree it for publication.

This manuscript is a resubmission of an earlier submission. The following is a list of the peer review reports and author responses from that submission.

Round  1

Reviewer 1 Report

This article presents a calibration method for a flash LiDAR based on iTOF measurement, including delay based black-box calibration and lens-distortion calibration. My comments are following:

1.      In the text from line 51 to 97, the author introduces many different types of calibration and algorithms without comparison. Please highlight the cons and pros of these methods, and do a comparison with the method proposed in this paper.

2.      It is more popular to use term “CMOS photodetector array”, instead of “array CMOS photodetector”.

3.      In line 124, the author mentioned ‘the calibration method of Zhang Zhengyou are used to …’ without reference. Please add a reference.

4.      In line 127,”optimize the distortion” should be ‘reduce the distortion’

5.      The text in line 200 – 203 is very difficult to read, “Because the actual modulation wave is not exactly the same as the sinusoidal wave or rectangular wave and because the sinusoidal modulation and demodulation mode is simpler and more real-time than the rectangular modulation and demodulation mode, the actual flight time is usually calculated using equation (3)’. please re-organize the expression.

6.      I strongly suggest the authors to rewrite the English.

7.      The authors mentioned ‘black-box calibration’ several times in the text without any explanation what is a ‘black-box calibration’. Please briefly explain it at the introduction, so the audient can get a point at the beginning.

8.      The figure number gets incorrect starting from page 10.

9.      In table 1, please comment on the different between the first and third cases, since they look like very similar? Why D2 is used twice in each interpolation. I assume one of them should be D3 ?

10.   In page 12, from figure 6, the distance error is more than 2 m at tTOF of 10 ns. However for a tTOF =10ns, it corresponds to a distance of 1.5m. I doubt  on the simulation result of a 2m distance error at 1.5m distance, which is too large. Please comment on how the error is calculated and simulated ?

11.   Since the reflected signal intensity degrades exponentially with distance, did the authors take the signal intensity into account during the error simulation? If it does, please comment on how it affects the distance error ? If it doesn’t, please explain why it is ignored.

12.   Please comment on the ambient light condition of the measurement in Figure 19. How the ambient light affects the distance error and how long it takes for recording one frame?